# Effect of Depressive Disorders and Their Pharmacological Treatment during Pregnancy on Maternal and Neonatal Outcome

**DOI:** 10.3390/jcm11061486

**Published:** 2022-03-09

**Authors:** Giulia Parpinel, Gianluca Rosso, Arianna Galante, Chiara Germano, Elena Aragno, Flavia Girlando, Alessandro Messina, Maria Elena Laudani, Alessandro Rolfo, Rossella Attini, Alberto Revelli, Giuseppe Maina, Bianca Masturzo

**Affiliations:** 1Department of Surgical Sciences, University of Turin, 10126 Turin, Italy; giulia.parpinel@edu.unito.it (G.P.); arianna.galante@edu.unito.it (A.G.); girlandofl@gmail.com (F.G.); dralessandromessina@gmail.com (A.M.); melaudani@cittadellasalute.to.it (M.E.L.); alessandro.rolfo@unito.it (A.R.); rattini@cittadellasalute.to.it (R.A.); aerre99@yahoo.com (A.R.); bianca.masturzo@aslbi.piemonte.it (B.M.); 2Psychiatric Unit, San Luigi Gonzaga Hospital, Regione Gonzole 10, 10043 Orbassano, Italy; gianluca.rosso@unito.it (G.R.); giuseppe.maina@unito.it (G.M.); 3Department of Neurosciences “Rita Levi Montalcini”, University of Turin, Via Cherasco 15, 10126 Turin, Italy; elena.aragno@hotmail.it; 4Department of Obstetrics and Gynaecology, Infermi Hospital, 13875 Ponderano, Italy; 5Department of Obstetrics and Gynaecology 2U, Sant’Anna Hospital, University of Turin, 10126 Turin, Italy

**Keywords:** psychotropic drugs, antidepressants, cesarean section, pregnancy, depression

## Abstract

**Purpose:** Depressive disorders are related to obstetrical and neonatal complications. The purpose of this study is to evaluate the outcomes of pregnancy in women suffering from depressive disorders, who are treated or not treated with pharmacotherapy during pregnancy. **Methods:** The maternal and neonatal outcomes of 281 pregnant women with depressive disorders (D group—DG), who delivered their babies at Sant’Anna Hospital of Turin, were compared with those of a control group of 200 depression-free, healthy, pregnant women, who were matched for maternal age (C group—CG). Of the depressed patients, those who received pharmacotherapy during pregnancy (DG-Tr, *n* = 199, 70.8%) were compared with those who did not (DG-Untr, *n* = 82, 29.2%). The comparisons were performed using χ^2^ tests for categorical variables and ANOVA for continuous variables. A linear regression analysis was run to examine the association between APGAR scores at 5 min and certain clinical variables. **Results:** The women in DG showed higher rates of cesarean section, preterm delivery, induction of labor and SGA babies, and low neonatal weights and 5-min APGAR scores, compared to the untreated patients. Those treated with psychotropic drugs showed lower rates of cesarean section, but lower 5-min APGAR scores, compared to those who were untreated. However, after controlling for confounding variables, the 5-min APGAR scores were linearly associated with neonatal weight and not with the use of psychotropic treatment. No significant differences were observed between the treated and untreated women, regarding the rates of preterm delivery, induction of labor, SGA and low neonatal weight. **Conclusion:** In pregnant patients with depressive disorders, poorer outcomes are expected vs. healthy controls. Pharmacological treatment is associated with a reduced rate of cesarean section, without inducing other complications for the mother and the newborn.

## 1. Introduction

Pregnancy and puerperium are considered high-risk time periods for the onset or recurrence of depression [1]. Of all pregnant women, up to 70% complain of depressive symptoms during pregnancy, and 10–16% of them fulfill the criteria of true depressive disorders [2,3,4,5]. A depressive disorder during the peri-partum period can interfere with normal feelings of motherhood and newborn care [6]. These women need greater attention and multi-disciplinary care [7].

The decision to treat depressive disorders during pregnancy with psychotropic medications should be carefully considered, weighing the risk of prenatal exposure to drugs vs. the risk, for both mother and child, of the consequences of maternal dees pression [8,9,10]. Antidepressant therapy should not be discontinued during pregnancy, but should be modified by choosing only one drug at the lowest effective dosage. Among antidepressants, SSRIs (selective serotonin reuptake inhibitors) have lower risks than other antidepressants, and fluoxetine is the safest among them. A slight increase in fetal toxicity is reported for tricyclic antidepressants, such as amitriptyline, imipramine and nortriptyline, while paroxetine, taken in the first trimester, may be associated with fetal heart defects, and venlafaxine with an increased risk of high blood pressure at high doses [11,12,13].

Psychotropic drugs, taken during pregnancy, may be responsible for congenital malformations (teratogenic risk), neonatal toxicity (withdrawal symptoms) and/or long-term neuro-behavioral disorders [14,15,16,17,18]. On the other hand, untreated depressive disorders may also have the following important consequences [19]: deterioration of the general health of the woman, poor adherence to obstetrical checks, increased risk of toxic abuse, smoke, malnutrition and sleep disorders. These conditions, in turn, may increase some fetal and neonatal risks (e.g., miscarriage, preterm birth, low birth weight, intrauterine growth restriction, and psychiatric and behavioral disorders after birth) [1,10,14,20].

The importance of distinguishing the effects of maternal depression from the effects of psycho-pharmacological treatment was highlighted in previous studies, but, so far, no study has clearly established which is the impact of maternal disease and which is that of medications during the obstetrical evolution of the pregnancy. Using over ten years of clinical data, the aim of the present study is to evaluate the outcomes of pregnancy and delivery, as well as newborns’ health, in women suffering from depressive disorders, who were treated or not treated with pharmacotherapy during gestation, distinguishing the effects of the disease from those of its treatment.

## 2. Methods

### 2.1. Patients

Pregnant women with depressive disorders referred to Sant’Anna Hospital (Regional Reference Center for Psychiatric Diseases in Pregnancy) between 2007 and 2018 were considered. Among 704 pregnant women with depressive disorders in our database, we excluded those who aborted (*n* = 89), delivered in other hospitals (*n* = 152), with other concomitant severe medical conditions (*n* = 15), with an associated psychiatric disorder other than depression (*n* = 82), with discontinuous psycho-pharmacological treatment during pregnancy (*n* = 83) or with exclusive non-pharmacological therapies (*n* = 3). Finally, a data analysis was performed on 281 pregnant women with depressive disorders (depressed patients group—DG).

Privacy-sensitive personal data were collected and encrypted. The following demographic and clinical data were recorded and analyzed: age, parity, smoking habit (>4 cigarettes/day), body mass index (BMI), type of depressive disorder, type of psycho-pharmacological treatment, gestational age at delivery, induction of labor, mode of delivery, neonatal weight and APGAR score 5 min after birth. A newborn with weight < 10th centile for gestational age was defined as small for their gestational age (SGA) [21]. The diagnosis of depressive disorder was certified by a psychiatrist according to the Diagnostic and Statistical Manual of Mental Disorders–Fifth Edition (DSM-5) [22].

The DG group was compared with a control group (CG), matched for maternal age, which was composed of 200 women with no mental illness and physiological pregnancy, admitted to Sant’Anna Hospital in the same time period, matching for parity, gestational age and BMI. Further, the DG was stratified into the following two subgroups: (a) patients who were treated with pharmacological therapy during gestation (DG-Tr), and (b) patients who did not take any psychotropic medication during pregnancy (DG-Untr, including women who interrupted pre-existing treatment before starting pregnancy).

### 2.2. Statistical Analysis

The normality of data distribution was evaluated by using Shapiro–Wilk and Kolmogorov–Smirnov tests. Demographic and clinical features were expressed as mean ± standard deviation (SD) for continuous variables, and as frequencies and percentages for categorical variables. The analysis compared (a) DG vs. CG, and (b) DG-Tr vs. DG-Untr. Comparisons were performed using χ^2^ tests for categorical variables and ANOVA for continuous variables. Furthermore, a linear regression model was used to identify variables associated with 5-min APGAR scores (the dependent variable). Results were expressed as 2-sided *p* values rounded to 3 decimal places; statistical significance was set at a *p* value < 0.05. All statistical analyses were performed by SPSS software (version 22.0).

## 3. Results

### 3.1. Depressed Patients (DG) vs. Controls (CG)

The comparison between pregnant patients with depressive disorders (DG) and healthy pregnant controls matched for maternal age (CG) is shown in Table 1. The smoking habit rate was significantly higher in DG, whereas the BMI and parity were comparable. The rates of preterm delivery, SGA, induced labor and cesarean section were significantly higher in the DG than among the controls, whereas neonatal weight and APGAR scores at 5 min post-partum were significantly lower.

### 3.2. Depressed Patients Receiving Pharmacological Treatment (DG-Tr) vs. Those Untreated (DG-Untr)

The comparison between the group of pregnant patients with depressive disorders receiving medications (DG-Tr) and those untreated (DG-Untr) is shown in Table 2. Within DG-Tr, 122 women (61.3%) received monotherapy and 77 women (38.7%) received polytherapy (Figure 1). The untreated patients were significantly younger than those in DG-Tr. We observed a significantly higher percentage of cesarean sections in DG-Untr. Conversely, the APGAR scores at 5 min were slightly, but significantly, lower in DG-Tr. No differences were observed in parity, smoking habits, BMI, gestational age at delivery, rate of preterm delivery, rate of labor induction, SGA and neonatal weight.

A linear regression model was developed, setting the APGAR score at 5 min after birth as the dependent variable; the independent variables were age, smoking habit, pharmacological treatment, cesarean section, neonatal weight and preterm birth. The linear regression analysis (Table 3) showed a significant linear association between APGAR scores and neonatal weight, but not between APGAR scores and pharmacological treatment.

## 4. Discussion

It is well established that women with depressive disorders who start a pregnancy require strict monitoring, aimed at the early recognition of any risky behaviors that could endanger the health of both the mother and neonate [12,13,23]. These patients are psychologically fragile, and have a lower ability to cope with stress, as well as lower compliance to obstetricians’ advice. In fact, comparing pregnant women with depressive disorders (DG) with healthy pregnant women, matched for maternal age (CG), we found that the former showed a significantly higher rate of smoking habits in pregnancy, suggesting that it was more difficult to obtain adequate cooperation between patients and obstetric teams. Further, it is well known that many women experience sleep alterations during pregnancy, especially in the last trimester [24]; for pregnant women complaining of depressive disorders, sleep alterations can lead to increased irritability and a further reduction in compliance to the physicians’ prescriptions.

The higher percentages of labor inductions and cesarean sections in DG could be explained by the greater fragility of these patients, who need to be reassured, as much as possible, about the duration of labor, and who have greater difficulty in coping with the pain related to uterine contractions. From this perspective, it is useful, after obtaining adequate information regarding the patient during the entire pregnancy, to program hospitalization at 38–39 weeks of gestation, in order to improve their compliance. In addition, the physician’s inexperience in dealing with psychologically unstable patients could have contributed to the increased cesarean section rate, although there is no psychiatric indication for choosing a cesarean section instead of a spontaneous delivery when no obstetrical complications occur [25,26].

Regarding neonatal conditions, the women with depressive disorders had higher rates of preterm deliveries, SGA babies, low neonatal weight and reduced APGAR scores compared to the healthy women. Indeed, depressive disorders and psychotropic drugs may both affect fetal development. On one hand, depressive disorders may lead to excessive stimulation of the hypothalamic–pituitary–adrenal axis (HPA) in response to stress, unhealthy behaviors and poor attendance for obstetric care [27]; on the other hand, antidepressants readily cross the placenta barrier, and may negatively affect fetal development [2,20,28].

In order to differentiate the effects of maternal depression itself from the effects of psychotropic drugs, we compared pregnant women with depressive disorders treated with medications vs. those untreated. From our study, we excluded the patients who discontinued the treatment during pregnancy to avoid bias related to the different times of interruption, and to the fact that the interruption could determine a modification of the obstetrical outcome. On the other hand, the choice of some patients to refuse the treatment is probably dictated by a fear of the teratogenic risks possibly connected to the therapy. Overall, approximately two thirds of the patients were under pharmacological treatment (DG-Tr, 70.8%, 61.3% in monotherapy), and one third were untreated (DG-Untr, 29.2%), reflecting the recommendations of the most recent guidelines [11,12], which encourage the treatment of depressed pregnant women when needed, preferably with a single medication. Overall, our data confirmed the following benefits of this policy: in patients affected by depressive disorders, the intake of psychotropic medication had a positive impact on their pregnancy outcome, leading to a significant reduction in the cesarean section rate, likely due to the better control of depressive symptoms, such as anxiety [29].

The possible risks of neonatal complications suspected to be related to psychotropic therapy in patients with depressive disorders (such as preterm birth, low birth weight and low APGAR scores) have been examined in our study. We only observed a significantly lower 5-min APGAR score in the treated group when compared to the untreated women. Among the independent variables entered in the linear regression analysis, only neonatal weight was found to be associated with the APGAR score at 5 min, which is consistent with previous studies [2,27], while there was no significant association between APGAR score and pharmacological therapy. In depressed pregnant women, the disease-linked response to stress could involve excessive stimulation of the HPA axis, with a cascade of events leading to both intrauterine growth restriction and a low APGAR score [20,30,31,32] independently from the effect of medications. Unfortunately, the lack of data on the circulating levels of the administered drugs prevents an understanding of whether a dose-dependent correlation between psychotropic treatment and a low APGAR score at 5 min exists. Indeed, a low APGAR score in treated depressed patients could also be linked to a mild and self-limiting neonatal adaptation syndrome, with, overall, little clinical relevance.

Taking into account both the need to achieve optimal compensation for the mental illness and the obvious issue of neonatal health, our data indicate that the effects of pharmacological therapy are less serious than the complications observed in patients who are uncompensated because of a lack of treatment. It is important to provide information regarding the use of psychotropic medication during pregnancy, in order to avoid the spontaneous discontinuation of treatment and the incorrect intake of medications, as well as adequate psychoeducation about the potential modifications of their mental disease. According to the literature [33,34,35], which demonstrates that careful and rational use of drugs in pregnant patients suffering from chronic diseases can improve pregnancy outcomes, the present study confirms this concept in the area of depressive disorders. Moreover, Pearson et al. [36] suggest that treating maternal antenatal depression could even prevent the children of those treated becoming depressed during adulthood.

We acknowledge that our study has some limitations that should be taken into account. Firstly, its retrospective nature does not allow the causal relationship between some variables to be ascertained, as potential confounders (e.g., circulating medication levels) were unknown. Secondly, the data collection finished in the immediate postpartum period, preventing us from being aware of the eventual late consequences of pharmacologic therapy on the offspring. Finally, as Sant’Anna Hospital is the reference center for psychiatric illnesses in pregnancy in its geographic area, it is possible that women with depressive disorders included in the study were more likely to be affected by illnesses that were more severe than average.

In conclusion, our study enrolled a numerically significant population, homogeneously admitted to the same obstetrical center, demonstrating that depressed pregnant patients who do not receive pharmacological treatment have a higher risk of pregnancy complications and worse clinical outcomes than those who are treated. Therefore, we emphasize the need to choose appropriate psychotropic therapies to ensure the best management of women suffering from depressive disorders during pregnancy and the peri-partum period, and to inform patients with depressive disorders and their relatives about the benefits obtained from medical treatment during pregnancy.

## Figures and Tables

**Figure 1 jcm-11-01486-f001:**
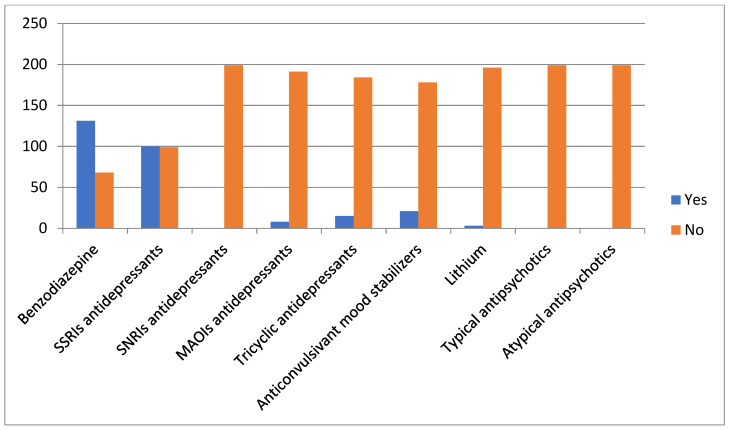
Type of psychopharmacological treatments among pregnant women with pharmacologically treated depressive disorders (DG-Tr, *n* = 199). SSRIs: selective serotonin reuptake inhibitors; SNRI: serotonin-norepinephrine selective reuptake inhibitors; MAOIs: monoamine oxidase inhibitors.

**Table 1 jcm-11-01486-t001:** Comparison of demographic and clinical variables of the group of pregnant women with depressive disorders (DG, *n* = 281) vs. the group of matched controls with physiological pregnancy (CG, *n* = 200).

	DG (*n* = 281)	CG (*n* = 200)	χ^2^/F	*p*
BMI	27.20 ± 4.30	25.80 ± 7.60	0.979	0.373
Parity, *n* (%)				
• nulliparous	139 (49.50)	108 (54.00)	0.961	0.187
• multiparous	142 (50.50)	92 (46.00)		
Smoking habit, *n* (%)				
• yes	91 (32.40)	1 (0.50)	76.789	<0.001
• no	190 (67.60)	199 (99.50)		
Gestational age at delivery (weeks)	40.00 ± 2.90	39.16 ± 1.70	0.713	0.562
Preterm birth, *n* (%)				
• yes	29 (10.40)	7 (3.50)	7.907	0.003
• no	251 (89.60)	193 (96.50)		
Induction of labor, *n* (%)				
• yes	122 (43.40)	11 (5.50)	83.966	<0.001
• no	159 (56.60)	189 (94.50)		
Delivery, *n* (%)				
• spontaneous	134 (47.70)	169 (84.50)	67.924	<0.001
• cesarean section	147 (52.30)	31 (15.50)		
Neonatal weight (grams)	3017 ± 524.19	3259 ± 507.47	25.499	<0.001
SGA, *n* (%)				
• yes	37 (13.20)	11 (5.50)	7.646	0.004
• no	244 (86.80)	189 (94.50)		
APGAR score at 5 min	8.79 ± 0.66	8.99 ± 0.29	16.163	<0.001

BMI: body mass index, SGA: small for gestational age.

**Table 2 jcm-11-01486-t002:** Comparison of demographic and clinical variables of the group of pregnant women with depressive disorders receiving pharmacological treatment (DG-Tr, *n* = 199) vs. the group of untreated pregnant women with depressive disorders (DG-Untr, *n* = 82).

	DG-Tr (*n* = 199)	DG-Untr (*n* = 82)	χ^2^/F	*p*
Age (years)	35.04 (±4.890)	33.34 (±5.51)	6.499	0.011
BMI, mean ± SD	26.30 ± 6.70	26.50 (±6.90)	0.876	0.395
Parity, *n* (%)				
• nulliparous	111 (50.80)	38 (46.30)	0.452	0.501
• multiparous	98 (49.20)	44 (53.70)		
Smoking habit, *n* (%)				
• yes	61 (30.70)	30 (36.60)	0.933	0.334
• no	138 (69.30)	52 (63.40)		
Gestational age at delivery (weeks)	40.00 ± 3.02	38.94 ± 2.59	0.320	0.572
Preterm birth, *n* (%)				
• yes	20 (10.10)	9 (11.00)	0.048	0.827
• no	178 (89.90)	73 (89.00)		
Induction of labor, *n* (%)				
• yes	102 (51.30)	62 (75.60)	17.062	0.072
• no	97 (48.70)	20 (24.40)		
Delivery, *n* (%)				
• spontaneous	104 (52.30)	30 (36.60)	5.720	0.017
• cesarean section	95 (52.30)	52 (63.40)		
Neonatal weight (grams)	3025.72 ± 523.46	2996.28 ± 528.58	0.182	0.670
SGA, *n* (%)			0.218	0.641
• yes	25 (12.60)	12 (14.60)		
• no	174 (87.40)	70 (85.40)		
APGAR score at 5 min	8.69 ± 0.96	8.93 ± 0.49	4.638	0.032

BMI: body mass index, SGA: small for gestational age.

**Table 3 jcm-11-01486-t003:** Linear regression analysis of clinical variables potentially affecting the APGAR score 5 min after birth.

	Unstandardized Coefficient		Standardized Coefficient			
	B	Standard Error	Beta	t	*p*	% CI
Psychiatric treatment	−0.178	0.121	−0.096	−1.467	0.144	−0.417–0.061
Cesarean section	−0.320	0.212	−0.188	−1.508	0.133	−0.738–0.098
Neonatal weight	0.000	0.000	0.213	3.488	0.001	0.000–0.001
Preterm birth	0.306	0.192	0.109	1.591	0.113	−0.73–0.685
Maternal age	0.006	0.010	0.037	0.618	0.537	−0.13–0.026
Smoking habit	−0.142	0.154	−0.078	−0.921	0.358	−0.446–0.162

## Data Availability

The data presented in this study are available on request from the corresponding author. The data are not publicly available due to privacy policies.

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
