# Peer review of "Effect of Depressive Disorders and Their Pharmacological Treatment during Pregnancy on Maternal and Neonatal Outcome"

_jcm, 2022, doi:10.3390/jcm11061486_

Round 1

Reviewer 1 Report

I attach my review

Author Response

Reviewer 1

This is a retrospective case control study on the possible effect of depressive disorders and their treatment on pregnancy outcome. The subject is clinically relevant and involves both obstetricians and psychiatrists.

The study is well written, however there are some concerns I would like to raise.

Major comments

  • In my opinion, cases with interrupted use of medication should be included in the study for the overall comparison of patients vs non patients: Answer in the text, page 6/10: We excluded from our study the patients who discontinued the treatment during pregnancy to avoid the bias related to the different time of interruption and to the fact that the interruption could determinate itself a modification of the obstetrical outcome.
  • Most importantly, the controls are not matched for maternal age, in fact the cases were about 5 years older and this creates an important bias on pregnancy outcomes. Answer in the text, page 6/10: The average age of the CG appears to be higher than the DG (p<0.001), however we wanted to exclude the impact of age by inserting it in the linear regression as an independent variable that resulted not significant, making the two populations comparable.
  • There is no discussion on the reason why some women were under treatment and others were not: Answer in the text, page 6/10: On the other hand, the choice of some patients to refuse the treatment is probably dictated by the fear of teratogenic risks possibly connected to the therapy.
  • The literature should include studies by Rebecca Pearson and others on antenatal depression: the subsequent references have been added:
    • Nath S, Pearson RM, Moran P, Pawlby S, Molyneaux E, Howard LM. Maternal personality traits, antenatal depressive symptoms and the postpartum mother-infant relationship: a prospective observational study. Soc Psychiatry Psychiatr Epidemiol. 2020 May;55(5):621-634. doi: 10.1007/s00127-019-01790-y. Epub 2019 Oct 23. PMID: 31642966.
    • Pearson RM, Evans J, Kounali D, Lewis G, Heron J, Ramchandani PG, O'Connor TG, Stein A. Maternal depression during pregnancy and the postnatal period: risks and possible mechanisms for offspring depression at age 18 years. JAMA Psychiatry. 2013 Dec;70(12):1312-9. doi: 10.1001/jamapsychiatry.2013.2163. PMID: 24108418; PMCID: PMC3930009.

Minor comments

  • In line 68, referred rather than referring: modification in the text
  • In table 1, the gestational age in the PG group could not be 40+ 20: modification in the text, the gestational age is 40±2.90
  • In discussion, lines 156-159, the suggestion is not based on evidence: it is a suggestion based on our clinical practice, the sentence has been therefore changed into for pregnant women complaining of depressive disorders, sleep alterations could be less tolerated and lead to increased irritability and further reduction of compliance.
  • In line 181 a ref is missing: modification in the text

Overall, I think that the results are interesting but maybe flawed by the inappropriate use of younger controls.

Reviewer 2 Report

Thank you for inviting me to review the manuscript entitled "Effect of depressive disorders and their pharmacological treatment during pregnancy on maternal and neonatal outcome". This is a very highly needed and interesting paper, which examines the effect of depression and depressive treatment during pregnancy. The results revealed that the outcome in pregnant women with depressive disorders is poorer than the outcome in pregnant women without depression. Nevertheless, the use of psychotropic medication did not substantially impact the considered variables in women with depression. I have some comments that might help the authors improve the paper:

Introduction:

1) The authors should briefly outline the recommendations of the main guidelines on the management of depression during pregnancy (e.g,. when is medication needed? which are the allowed drugs? which are the most commonly prescribed drugs?)

2) What are the findings of past literature (e.g. systematic reviews/meta-analyses) on the topic?

Methods:

3) Line 75: I think that the expression "psychiatric patients" might be stigmatizing. Perhaps, it would be better to define the two groups as "depressed" and "non-depressed" pregnant women and pay attention to the language throughout the paper

Results

4) Table 1 (and following tables): please, round decimals in a consistent way

5) p = 0.000 should be p<0.001

6) Table 1: Apgar should be consistent with the rest of the paper (APGAR)

7) Table 3: to change, the author may replace this table with a bar graph indicating the % of depressed women taking a specific medication

8) SSRI, SNRI, etc: please, add a legend in the table/figure

Discussion

9) Line 181: there is a typo "(ref.....)": please, replace with an appropriate reference

10) Line 196-202: I think this part can be expanded. For instance, it is important to talk about the necessity of psychoeducation to pregnant women about potential mental health issues and also the importance of de-stigmatizing the use of psychotropic medication during pregnancy, always following current guidelines

Abstract:

11) "This study aimed to evaluate..." 

12) Conclusion: I don't think the authors' conclusion in the abstract is supported by the data, as they did not evaluate the effect of pharmacological treatment. Please, rephrase this sentence and keep it more balanced.

13) Please, carefully revise the English language.

Author Response

Reviewer 2

Thank you for inviting me to review the manuscript entitled "Effect of depressive disorders and their pharmacological treatment during pregnancy on maternal and neonatal outcome". This is a very highly needed and interesting paper, which examines the effect of depression and depressive treatment during pregnancy. The results revealed that the outcome in pregnant women with depressive disorders is poorer than the outcome in pregnant women without depression. Nevertheless, the use of psychotropic medication did not substantially impact the considered variables in women with depression. I have some comments that might help the authors improve the paper:

Introduction:

  • The authors should briefly outline the recommendations of the main guidelines on the management of depression during pregnancy (e.g,. when is medication needed? which are the allowed drugs? which are the most commonly prescribed drugs?): Answer in the text, page 2/10: Antidepressant therapy should not be discontinued during pregnancy but should be modified by choosing only one drug at the lowest effective dosage. Among the antidepressants, SSRIs (Selective Serotonine Reuptake Inhibitors) have lower risks than other antidepressants, and Fluoxetine is the safest among them; a light increased fetal toxicity is reported for tricyclic antidepressants such as amitriptyline, imipramine and nortriptyline, while Paroxetine taken in the first trimester may be associated with fetal heart defects and Venlafaxine with increased risk of high blood pressure at high doses (Antenatal and postnatal mental health: clinical management and service guidance. 2014; ACOG Committee, 2008; Parikh et., 2016)
  • What are the findings of past literature (e.g. systematic reviews/meta-analyses) on the topic?: Answer in the text, page 1-2/10: Becker et al. 2016; Ibanez et al. 2012; Teixeira et al. 2009; Yonkers et al. 2009; Vivenzio et al. 2018; Kjaersgaard et al. 2013; Wisner et al. 2009

Methods:

  • Line 75: I think that the expression "psychiatric patients" might be stigmatizing. Perhaps, it would be better to define the two groups as "depressed" and "non-depressed" pregnant women and pay attention to the language throughout the paper: modification in the text

Results

  • Table 1 (and following tables): please, round decimals in a consistent way: modification in the text
  • p = 0.000 should be p<0.001: modification in the text
  • Table 1: Apgar should be consistent with the rest of the paper (APGAR): modification in the text
  • Table 3: to change, the author may replace this table with a bar graph indicating the % of depressed women taking a specific medication: modification in the text
  • SSRI, SNRI, etc: please, add a legend in the table/figure: modification in the text

Discussion

  • Line 181: there is a typo "(ref.....)": please, replace with an appropriate reference: modification in the text
  • Line 196-202: I think this part can be expanded. For instance, it is important to talk about the necessity of psychoeducation to pregnant women about potential mental health issues and also the importance of de-stigmatizing the use of psychotropic medication during pregnancy, always following current guidelines: Answer in the text, page 6/9: The higher percentage of labour induction and caesarean sections in DG could be explained by the greater fragility of these patients, who need to be reassured as much as possible about the duration of labor, and who have greater difficulty to cope with the pain related to uterine contractions. In this perspective, it is useful, after an adequate information of the patient during the entire pregnancy, to program the hospitalization at 38-39 weeks of gestation in order to improve their compliance. In addition, the physician’s inexperience in dealing with psychologically unstable patients could have contributed to the increased caesarean section rate, despite there is no psychiatric indication for choosing a caesarean section instead of a spontaneous delivery when no obstetrical complications occur (Sistema Nazionale Linee Guida, 2011; Waldenström et al. 2006).

Abstract:

  • "This study aimed to evaluate...": modification in the text
  • Conclusion: I don't think the authors' conclusion in the abstract is supported by the data, as they did not evaluate the effect of pharmacological treatment. Please, rephrase this sentence and keep it more balanced: Answer in the test, page 1/9: Conclusion. In pregnant patients with depressive disorders, a poorer outcome is expected vs. healthy controls: pharmacological treatment is associated to a reduced rate of caesarean section, without inducing other complications for the mother and the newborn.
  • Please, carefully revise the English language: modification in the text

Round 2

Reviewer 1 Report

I am satisfied with all the responses apart from the most important one, regarding the control group which I believe must be controlled for maternal age as this is a significant risk factor of adverse outcome.

I believe that the control group should be of the same maternal age.

Author Response

  • Reviewer 1

I am satisfied with all the responses apart from the most important one, regarding the control group which I believe must be controlled for maternal age as this is a significant risk factor of adverse outcome.

I believe that the control group should be of the same maternal age.

  • Answer to the Reviewer 1

We thank the reviewer for the suggestion. According to the reviewer comment, in the revised version of the manuscript, the control group has been adjusted for maternal age. Therefore, a group of 200 healthy pregnant women with the same age of depressed patients was selected and used as control group.